# Differentiation between enamines and tautomerizable imines in the oxidation reaction with TEMPO

Xiaoming Jie[1], Yaping Shang[1], Zhe-Ning Chen [1], Xiaofeng Zhang[1], Wei Zhuang[1] & Weiping Su[1]

Enamine and imine represent two of the most common reaction intermediates in syntheses, and the imine intermediates containing $\alpha$-hydrogen often exhibit the similar reactivity to enamines due to their rapid tautomerization to enamine tautomers. Herein, we report that the minor structural difference between the enamine and the enamine tautomer derived from imine tautomerization results in the different chemo- and regioselectivity in the reaction of cyclohexanones, amines and TEMPO: the reaction of primary amines furnishes the formal oxygen 1,2-migration product, $\alpha$-amino-enones, while the reaction of secondary amines under similar conditions generates exclusively arylamines via consecutive dehydrogenation on the cyclohexyl rings. The $^{18}$O-labeling experiment for $\alpha$-amino-enone formation revealed that TEMPO served as oxygen transfer reagent. Experimental and computational studies of reaction mechanisms revealed that the difference in chemo- and regioselectivity could be ascribed to the flexible imine-enamine tautomerization of the imine intermediate containing an $\alpha$-hydrogen.

[1] State Key Laboratory of Structural Chemistry, Center for Excellence in Molecular Synthesis, Fujian Institute of Research on the Structure of Matter, Chinese Academy of Sciences, 155 Yangqiao Road West, Fuzhou 350002, China. Correspondence and requests for materials should be addressed to W.Z. (email: wzhuang@fjirsm.ac.cn) or to W.S. (email: wpsu@fjirsm.ac.cn)

Enamines and imines are both key reactive intermediates of numerous classic organic reactions[1,2] and the recently developed aminocatalysis for transformations of aldehydes and ketones[3–6]. Due to the diverse reactivity of enamine intermediates, aminocatalysis not only accelerates α-functionalization of aldehydes and ketones with a variety of electrophiles by enhancing nucleophilicity of carbonyl α-carbon atoms[3–10], but also enables use of various nucleophiles or radical trapping reagents for α-functionalization[11–15] and even β-functionalization[16,17] of aldehydes and ketones via oxidation of enamines to generate delocalized radical cations or iminium ions. On the other hand, imines containing α-hydrogen can rapidly tautomerize to their enamine tautomers, and therefore often behave as enamines[18–22]. In this context, the tautomerizable imines have been reported to participate, through their enamine tautomers, in aldol-type condensation with aldehydes (Fig. 1a)[18], Michael addition to electrophilic olefins[19], Pd-catalyzed intramolecular α-C–H arylation of imines via imine α-palladation (Fig. 1b)[20,21], Cu-catalyzed aerobic oxygenation at the position α to imine group[22]. The similarity in reactivity between enamine and tautomerizable imine led us to consider what is the exact difference in reactivity between the enamine tautomer derived from tautomerizable imine and the real enamine. The effort to address this question would develop the chemistry of imine and enamine and gain an insight into their structure-reactivity relationship.

Recently, a series of α-aminoxylation reactions of carbonyl compounds[14,15,23–30] using 2,2,6,6-tetramethylpiperidine-1-oxyl (TEMPO) as oxidant has been established, some of which involved metal-promoted oxidation of the enamine intermediates generated in-situ from the condensation of aldehydes with secondary amines[14,15,23–28]. These elegant studies provided a good starting point for our investigation of the structure-reactivity relationship of enamine and tautomerizable imine.

Herein, we demonstrate the distinct difference between enamines and tautomerizable imines in the metal-free oxidation reactions with TEMPO as oxidant (Fig. 1c): the tautomerizable imines, which are generated in-situ from the condensation of cyclohexanones with primary amines, bring about oxygen transfer from TEMPO to α-carbon atom of imines and furnish α-amino enone products, whereas enamines generated in situ from the condensation of cyclohexanones with secondary amines undergo dehydrogenative aromatization to produce aryl amines.

## Results

**Discovery and development of α-amino enone formation reaction.** We started our investigation by examining the reaction of 4-tert-butyl-cyclohexanone (**1a**) with 4-chloro-aniline (**2a**) (1.5 equiv.) and TEMPO (1.5 equiv.), in which imine intermediate was expected to be generated in-situ from the condensation of cyclohexanone with aniline (Fig. 2). Initially, the reaction was carried out in the presence of Cu(OAc)$_2$ and 2,2′-bipyridine ligand (bpy), which is similar to the reaction conditions established for the α-aminoxylation of aldehydes with TEMPO via enamine intermediate[24]. Interestingly, the reaction of cyclohexanone with aniline produced a formal oxygen 1,2-migration compound, α-amino enone (**3a**), in 63% yield (entry 1) (the structure of this product was also established by single crystal X-ray diffraction analysis). After removing bpy ligand from the reaction system, this α-amino enone formation reaction obtained a higher yield (entry 2). Considering that the ketone-amine condensation to generate imine would be a prerequisite step in this reaction, we turned our attention to using Lewis acid catalyst for accelerating the ketone-amine condensation. As expected, the use of Lewis acid catalysts such as Yb(OTf)$_3$, Zn(OTf)$_2$ and AlCl$_3$ in place of Cu(OAc)$_2$ effected the desired reaction, albeit in lower yields (entries 3–5). 4-Methylanthranilic acid, which proved to be effective catalyst for ketone-amine condensation[31,32], could also promote α-amino enone formation in 37% yield (entry 6). Adding 3 Å molecular sieve (400 mg) to the reaction system, in conjunction with 10 mol% 4-methylanthranilic acid, significantly improved the reaction (77% yield) (entry 7), most likely because

Previous work :

**a** Similarity : nucleophilic addition of imine to aldehyde via imine tautomerization to enamine

**b** Similarity : intramolecular α-C-H arylation of imine via imine tautomerization to enamine

**This work :**

**c** Dissimilarity : α-oxygenation of imine by TEMPO versus consecutive dehydrogenation of enamine

**Fig. 1** Similarity and Dissimilarity in Reactivity between Enamine and Tautomerizable Imine. In both nucleophilic addition reaction (**a**) and Pd-catalysed intramolecular α-C-H arylation reaction of α-hydrogen-containing imine (**b**), imines participate in reaction through taotumerization to their enamine tautomers and show similar reactivity to enamine. The reaction of cyclohexanones, amines and TEMPO depends on the reaction intermediates and affords α-amino enones from primary amines and arylamines from secondary amines (**c**), exhibiting the dissimilarity in reactivity between enamine and α-hydrogen-containing imine

**Fig. 2** Optimization of α-amino-enone formation from primary amine[a]. [a] Reaction conditions: **1a** (0.3 mmol), **2a** (0.2 mmol), catalyst (10 mol%), solvent (1.0 mL), N2, 120 °C for 24 h. [b] Yields were determined by GC analysis using dodecane as an internal standard. [c] Conversion of TEMPO and yield of **5** were based on the amount of TEMPO and determined by GC analysis. [d] 15 mol% catalyst was used

| Entry | Catalyst (10 mol%) | Additive | Yield of **3a** (%)[b] | Conv. of TEMPO (%)[c] | Yield of **4** (%)[b] | Yield of **5** (%)[c] |
|---|---|---|---|---|---|---|
| 1 | Cu(OAc)2, bpy | - | 63 | 96 | trace | 94 |
| 2 | Cu(OAc)2 | - | 69 | 97 | trace | 94 |
| 3 | Yb(OTf)3 | - | 9 | 71 | 38 | 15 |
| 4 | Zn(OTf)2 | - | 10 | 71 | 34 | 17 |
| 5 | AlCl3 | - | 48 | 95 | trace | 73 |
| 6 | H3C—⬡(NH2)(CO2H) | - | 37 | 78 | 20 | 43 |
| 7 | H3C—⬡(NH2)(CO2H) | 3Å MS (400 mg) | 77 | 100 | 11 | 92 |
| 8[d] | H3C—⬡(NH2)(CO2H) | **3Å MS (400 mg)** | **86** | **100** | **trace** | **91** |

molecular sieve absorbed water generated in the ketone-amine condensation step to push the reversible ketone-amine condensation towards imine formation. Increasing loading of 4-methylanthranilic acid to 15 mol% further enhanced the yield to 86% (entry 8).

**Scope of α-amino enone formation reaction**. We next examined the scope of the metal-free method for facile syntheses of α-amino enones (Fig. 3). With respect to primary amine coupling partners, a broad range of primary amines served as competent substrates. A variety of anilines bearing electron-withdrawing (**3a, 3b, 3c, 3d, 3e, 3f, 3g**) and electron-donating (**3h, 3i**) groups afforded the corresponding products in generally good yields. The positions of substituents on the phenyl rings of anilines did not affect the reaction outcomes, as exemplified by 3-methyl aniline (**3j**) and 2-methyl aniline (**3k**). Similarly to anilines, both electron-deficient (**3n**) and electron-rich (**3l, 3 m**) heteroaromatic amines produced the corresponding products in good yields. Moreover, aliphatic primary amines (**3o, 3p, 3q**) including α-ester substituted amine (**3r**), also exhibited good performance in α-amino enone formation reactions. Meanwhile, a series of substituents on the ring of cyclohexanones, such as phenyl (**3s**), ester (**3t, 3u**) and Boc-protected amino (**3v**) groups, were tolerated in this reaction. This α-amino enone formation protocol was also compatible with the steric variation on the ring of cyclohexanones. In this regard, 4,4- and 3,3-dimethyl-substituted cyclohexanones (**3w, 3x**), polycyclic compounds containing cyclohexanone subunit (**3ad, 3ae**), and five-membered cyclic ketone (**3aa**) also smoothly underwent this reaction with good yields. Cyclohexanone lacking any substituent on the six-membered ring was suitable for this reaction to afford the desired product (**3ab**) in 70% yield. In

spite of a broad substrate scope with respect to cyclic ketone, α-substituted cyclohexanones did not participate in this reaction. Unfortunately, four-membered cyclic ketone did not work for this reaction either.

Our general method for synthesis of α-amino-enones from cyclohexanones and primary amines is attractive because traditional syntheses of α-amino-enones invoke multi-step procedures[33]. Due to their dual electronic attitude, α-amino-enones proved to be versatile building blocks in organic synthesis[33–36]. In this context, α-amino-enones have recently been used to prepare several important classes of heterocycles such as oxazines, azaspirones, quinolinones and quinolones in regio- and chemoselective fashion via controllable annulations[33]. Moreover, α-Amino-enones are amenable to intramolecular oxidative C-C bond formation to construct substituted tetrahydron-1H-carbazol-1-ones[34], a domino reaction with β-bromonitrostyrene to yield pyrroles[35], and intramolecular radical cyclization reaction to form bicycle N-heterocycles[36].

**Scope of dehydrogenative aromatization reaction of enamines**. When using secondary amines for α-amino enone formation, we found that the reaction of cyclohexanone with secondary amine mainly produced arylamine via consecutive dehydrogenation of cyclohexanone ring, with a trace of α-amino enone formed. The screening of reaction conditions did not change this product distribution (see Supplementary Table 2 for details). This metal-free secondary amine-induced consecutive dehydrogenation is interesting[37] since metal catalysts are generally required for dehydrogenative desaturation[38–42] and dehydrogenative aromatization[43–49] of carbonyl compounds. In this dehydrogenative aromatization, 4-methylanthranilic acid catalyst was observed to be detrimental to the reaction yield probably because it impeded

**Fig. 3** Reaction of cyclic ketones with primary amines[a]. [a]Reaction conditions: **1** (0.3 mmol), **2** (0.2 mmol), catalyst (15 mol%), TEMPO (0.3 mmol), 3 Å MS (400 mg), toluene (1.0 mL), N$_2$, 120 °C for 24 h. Isolated yields. [b] **1a** (0.4 mmol), **2m** (0.2 mmol), AlCl$_3$ (30 mol%), TEMPO (0.3 mmol), DCE (2.0 mL), N$_2$, 120 °C for 24 h. [c] **1a** (0.5 mmol) was used

the condensation of secondary imine with ketone. The condensation of primary amine with ketone catalyzed by anthranilic acid involves the initial condensation of anthranilic acid with ketone to form imine intermediate, followed by subsequent reaction with primary amine to afford the final imine product[31], while relatively bulky secondary amine was difficult to react with the imine intermediate generated from anthranilic acid and

ketone. Increasing the amount of TEMPO to 2.5 equivalents allowed the reaction of cyclohexanone with morpholine to furnish aryl amine **7a** in 62% yield in the absence of 4-methylanthranilic acid (Fig. 4). Cyclohexanones bearing a variety of functional group at 4-positions of their rings, such as, *tert*-butyl (**7c**, **7d**), phenyl (**7e**), ester (**7h**, **7i**) and amide groups (**7j**, **7k**) reacted with morpholine or thiomorpholine and TEMPO to

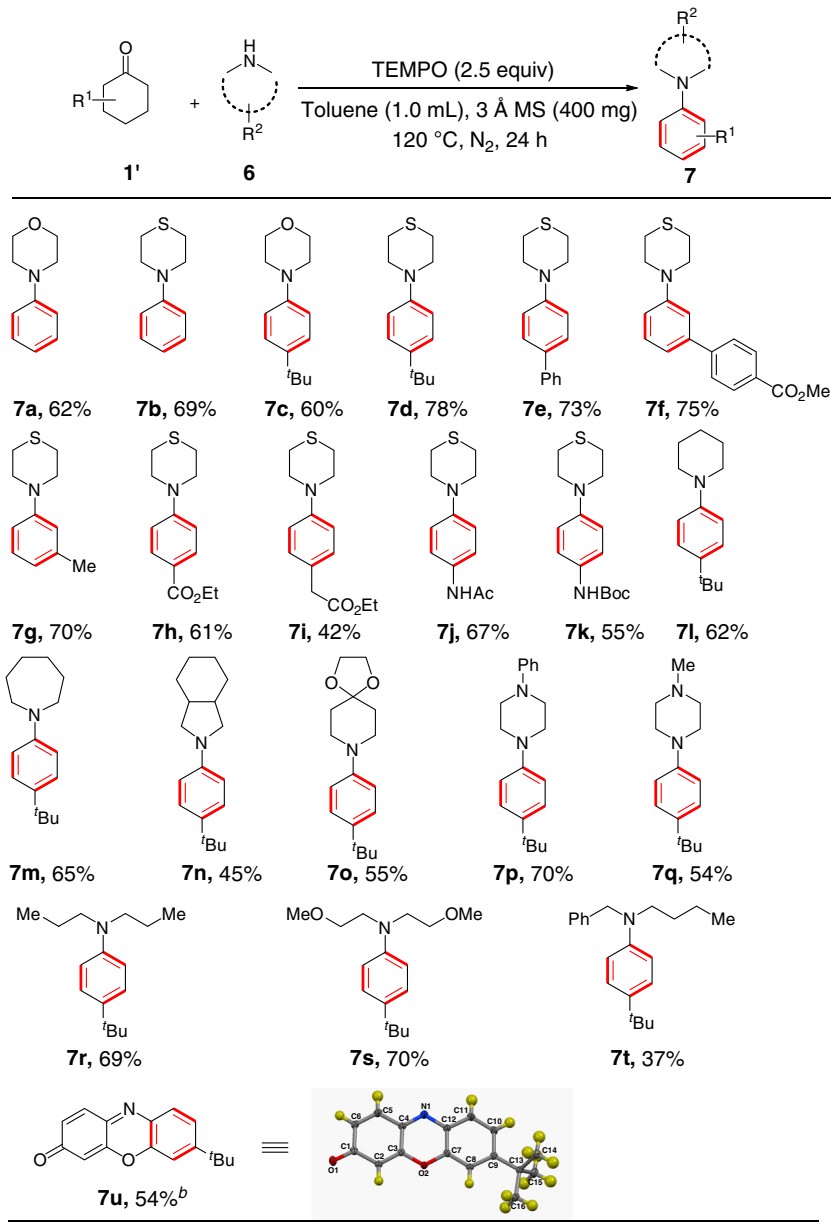

**Fig. 4** Reaction of cyclic ketones with secondary amines[a]. [a]Reaction conditions: **1'** (0.4 mmol), **6** (0.6 mmol), TEMPO (1.0 mmol), 3 Å MS (800 mg), toluene (1.0 mL), N₂, 120 °C for 24 h. Isolated yields. [b] TEMPO (2.4 mmol) and 3 Å MS (1000 mg) were used

produce the corresponding dehydrogenative aromatization products in good yields. Cyclohexanones (**7f**, **7g**) with substituents at 3-positions at their rings also worked well. Structurally diverse secondary amines served as competent coupling partners for efficient dehydrogenative aromatization whether they were cyclic (**7l**, **7 m**, **7n**, **7o**, **7p**, **7q**) or acyclic (**7r**, **7s**, **7t**) amines.

Given that enamines underwent the dehydrogenative aromatization reaction, the α-amino enone products from the reaction of cyclic ketones with primary amines could undergo dehydrogenative aromatization in the presence of excessive TEMPO. Indeed, the reaction of 4-*tert*-butyl-cyclohexanone with 4-chloroaniline in the presence of six equivalents of TEMPO gave an interesting 7-(*tert*-butyl)−3H-phenoxazin-3-one product in 54% yield (**7 u**, Fig. 4) (7u was also characterized by X-ray single-crystal diffraction analysis).

Our dehydrogenative aromatization to construct aryl amines from cyclohexanones and secondary amines provides a useful complement to Buchwald-Hartwig coupling[50,51] and Ullmann-

type aminations[52], and importantly open up an avenue to rapid synthesis of aryl amines from readily available starting materials.

**Mechanistic investigation**. We carried out the mechanistic investigations to gain a better understanding of the factors that control the chemical selectivity in the reaction of cyclohexanone with amine and TEMPO (Fig. 5). Treatment of the pre-synthesized imine (**9**) and enamine (**10**) with TEMPO under standard reaction conditions led to formations of α-amino enone (**3ac**) and arylamine (**7a**) (eqs. 1–2, Fig. 5), respectively, implicating that imine and enamine would be reaction intermediates in the reactions in question. The reactions of TEMPO with cyclo-hexanones and primary amines conducted at the lowered temperature (40 °C) produced α-aminoxylated ketone (**11**) (eq. 3, Fig. 5). This α-aminoxylated ketone likely resulted from the acid-promoted hydrolysis of α-aminoxylated imine in work-up step, which was supported by the observation that the reaction of cyclic

**Fig. 5** Experimental studies of mechanism. α-Hydrogen-containing imine reacts with TEMPO to form α-amino enone (1). Enamine reacts with TEMPO to form arylamine (2). The reaction of cyclohexanones, amines and TEMPO at 40 °C gives α-aminoxylated ketone (3). The $^{18}$O-labeling experiment shows that the oxygen atom of ketone group in the α-amino enone product originates from TEMPO (4)

ketone with TEMPO did not give any α-aminoxylated ketone in the absence of primary amine under otherwise identical conditions. $^{18}$O-labeling experiment revealed that the oxygen atom of ketone group in the α-amino enone product ($^{18}$O-3a) originated from TEMPO (eq. 4, Fig. 5), which, to the best of our knowledge, represents the first example of TEMPO serving as oxygen transfer reagent. The result from this $^{18}$O-labeling experiment revealed that the reaction of tautomerizable imine with TEMPO occur at α-carbon of imine. Moreover, the $^{18}$O-labeling experiment, in conjunction with the α-aminoxylation reaction at eq. 3 and MacMillan's report[24] on the metal-catalyst-free enamine catalysis for aminoxylation of aldehyde via enamine intermediate, implicated that the tautomerizable imine intermediate might react with TEMPO in the form of enamine to generate α-aminoxylated imine, which would be a key step lying in the reaction pathway to α-amino enones.

To understand the different selectivity for enamines and imines oxidation at molecular level, we conducted the density functional theory (DFT) calculations on the mechanism of oxidation reaction for model enamine molecule **A** and imine molecule **B** (see Figs. 6 and 3, Supplementary Figs. 8 and 9). Note that the enamine molecule **A** and imine molecule **B**, which both derive from aliphatic amines, were chosen to disclose the intrinsic difference between enamines and imines. To figure out whether substituents have impact on the reaction mechanism or not, reaction of imine generated from aniline was also calculated, which showed that this phenyl-substituted imine followed the similar reaction pathway to alkyl-substituted imine (Supplementary Fig. 10).

Based on the aforementioned α-aminoxylated ketone formation (eq. 3, Fig. 5), we reasoned that oxidation of enamine and tautomerizable imine by TEMPO would involve carbon-centered radical intermediate. The hydrogen abstraction in all possible reactive sites was thus calculated at first. As shown in Fig. 6a, β-site of enamine molecule **A** shows most active for C-H bond homolytic cleavage as compared to other carbon sites, which should be attributed to forming a relatively stable 5π-electron β-enaminyl radical intermediate[16]. The free energy profiles for enamine oxidation were displayed in Fig. 6b. We suggest that the proper proton-relay chain needs to be introduced when necessary for reducing the intrinsic reorganization energy of some transition states (Supplementary Fig. 7). Given that a certain amount of water can be generated from the condensation of ketone with amine and that a trace amount of water may come from reagents, the existence of catalytic amount of water is conceivable. Consequently, water was considered as a possible proton-relay chain here. A cluster made of three water molecules $(H_2O)_3$ was used in calculations, since it was indicated that $(H_2O)_3$ cluster represents a good compromise between a realistic description and system size for proton exchange processes[53,54]. As shown in Fig. 6b, hydrogen atom abstraction from β-carbon of enamine by TEMPO acts as the initial step for the dehydrogenative aromatization reaction to form a radical intermediate **LM1A**, which requires a free energy barrier of 33.2 kcal mol$^{-1}$ (**TS1A**). Then, this resultant radical intermediate rebounds with another TEMPO molecule, leading to formation of β-aminoxylated intermediate **LM2A**. Two different hydrogens exist in **LM2A**: one bonds to the carbon atom of the newly generated C–O bond

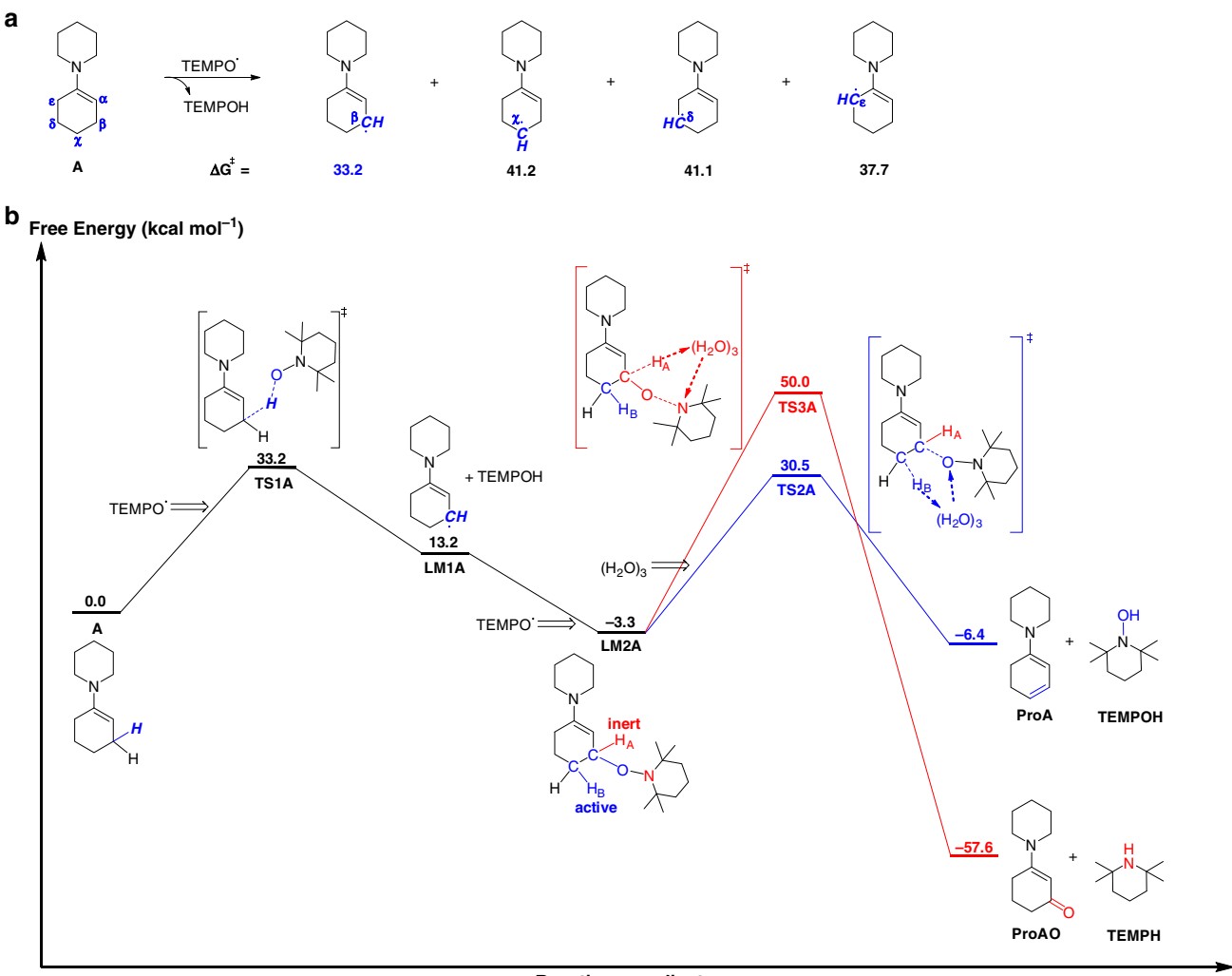

**Fig. 6** Computational investigation on the mechanism for oxidation of enamine by TEMPO. Numbers are Gibbs free energies (kcal mol⁻¹) with respect to the enamine reagent **A** and TEMPO. **a** Activation free energies of the hydrogen abstraction of enamine molecule **A** by TEMPO in different reactive sites. **b** Free energy profiles for oxidation of enamine molecule **A** by TEMPO. The pathway for the elimination of $H_B$/TEMPO is shown in blue line and the pathway for the elimination of $H_A$/piperdine is shown in red line

(denoted as $H_A$), and the other one bonds to the carbon atom adjacent to that C–O bond (denoted as $H_B$). Elimination of $H_A$/piperidine produces $\beta$-amino enone (**ProAO**) while eliminating $H_B$/TEMPO leads to amino diene intermediate (**ProA**) that is a precursor to aryl amine product. Our calculations show the elimination of $H_B$/TEMPO is feasible with a free energy barrier of 33.8 kcal mol⁻¹ (**TS2A**) with respect to the $\beta$-aminoxylated intermediate **LM2A**. However, elimination of $H_A$/piperidine requires an extremely high barrier of 53.3 kcal mol⁻¹ (**TS3A**). Therefore, although forming $\beta$-amino enone molecule **ProAO** is much energetically exothermic, formation of $\beta$-amino enone is unfeasible in kinetics due to the high barrier for the elimination of $H_A$/piperidine. As such, computational studies elucidate the selectivity of enamine reaction towards arylamine.

Interestingly, as displayed in Fig. 7a, the imine molecule **B** seems inert due to the relatively high barrier for hydrogen abstraction from all possible reactive sites. However, the enamine tautomer **LM1B** exhibits the desired activity in N-H bond as compared to its tautomer **B**, implicating the significance of imine-enamine tautomerization for activation of imine species. In addition, the activation of $\alpha$-C-H bond becomes most feasible as the result of imine-enamine tautomerization, indicating a

regioselectivity different from that of enamine molecule **A**, which is in agreement with the $\alpha$-aminoxylated ketone (**11**) identified from the reaction of TEMPO with cyclohexanone and primary amine (eq. 3, Fig. 5). As shown in Fig. 7b, in oxidation of $\alpha$-hydrogen-containing imine by TEMPO, reaction starts with an imine-enamine tautomerization via **TS1B** to form enamine species **LM1B** rather than the direct hydrogen abstraction from C-H bond (see Fig. 3a, b). The hydrogen abstraction of **LM1B** occurs subsequently from nitrogen by TEMPO. Due to formation of relatively active N-H bond, tautomerizable imine **B** shows higher activity towards hydrogen abstraction process than enamine **A**, reducing the activation free energy of this step to 31.5 kcal mol⁻¹ (**TS2B**). This imine-enamine tautomerization thus enhances the activity of imine **B** towards hydrogen abstraction, and leads to the regioselectivity for its $\alpha$-carbon site. As shown in Fig. 7b, the C-$H_A$ bond seems still inert for direct $H_A$/piperidine elimination, which requires a high barrier of 43.5 kcal mol⁻¹ (**TS3B'**). Alternatively, this inert C-$H_A$ bond can be activated via an imine-enamine tautomerization process, where the $\alpha$-aminoxylated enamine intermediate **LM4B** is formed with a feasible activation free energy of 33.0 kcal mol⁻¹ (**TS3B**). As a result, the inert C-$H_A$

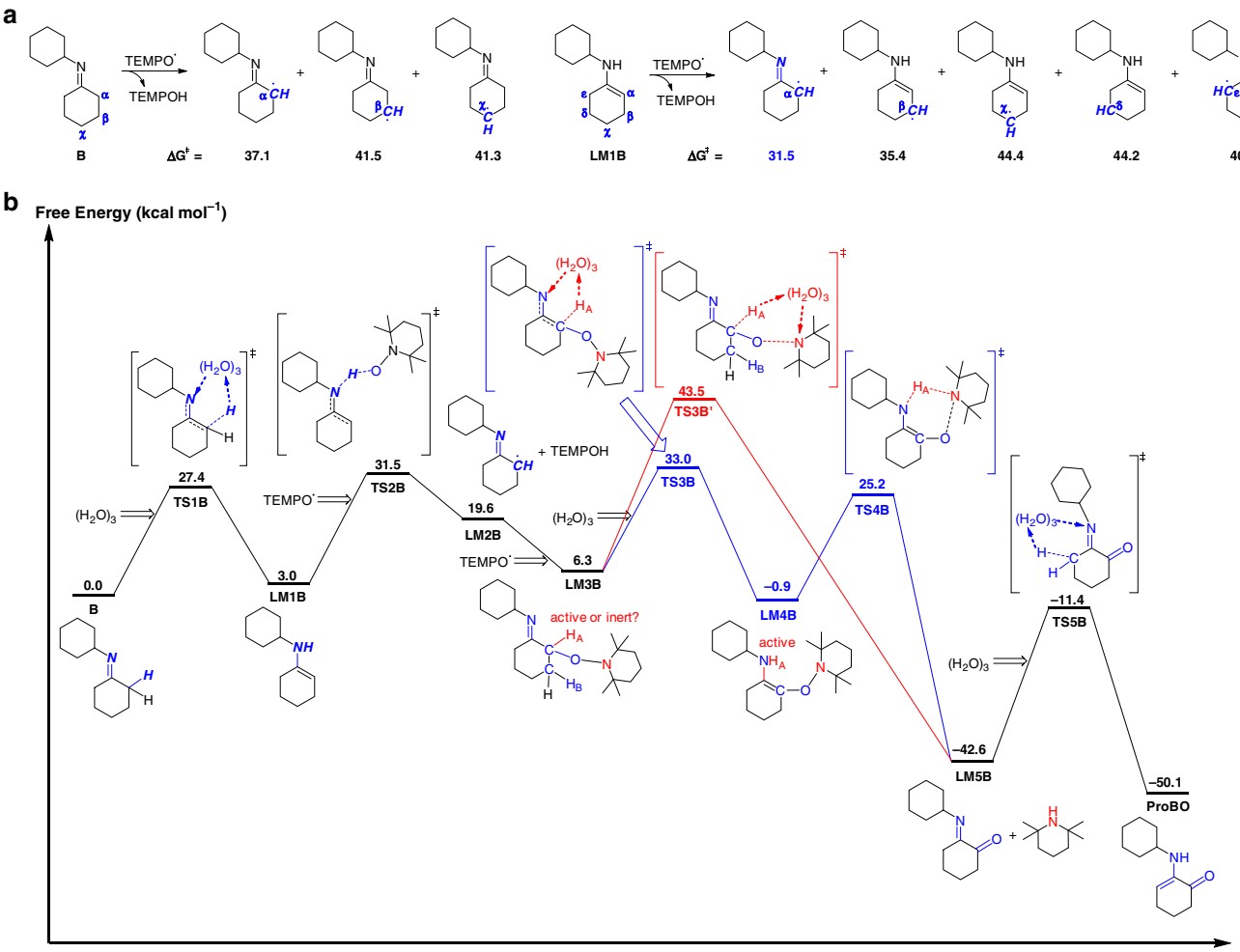

**Fig. 7** Computational investigation on the mechanism for oxidation of imine by TEMPO. Numbers are Gibbs free energies (kcal mol⁻¹) with respect to the imine reagent **B** and TEMPO. **a** Activation free energies of the hydrogen abstraction of imine molecule **B** and its tautomer **LM1B** by TEMPO in different reactive sites. **b** Free energy profiles for oxidation of imine molecule **B** by TEMPO. The pathway for the elimination of H$_A$/piperidine via imine-enamine tautomerization is shown in blue line and the pathway for the direct elimination of H$_A$/piperidine is shown in red line

bond in **LM3B** becomes active N-H$_A$ bond in **LM4B** for the 1,4-elimination of H$_A$/piperidine with an activation free energy of 25.2 kcal mol⁻¹ (**TS4B**). With the final tautomerization step via **TS5B**, the reaction channel to α-amino-enones becomes passable. The H$_A$/piperidine elimination channel to α-amino enone was further revealed to be more kinetically favorable than the H$_B$/TEMPO elimination channel to arylamine in the case of α-hydrogen-containing imine (see Supplementary Fig. 9), rationalizing the chemoselectivity of imine oxidation towards α-amino enone.

## Discussion

We have demonstrated that the regio- and chemo-selectivity in the reaction of cyclohexanone with amine and TEMPO heavily depends on the nature of the amines used: the reactions of primary amines produced α-amino-enone products via α-oxygenation of the imine intermediate while the reactions of secondary amines led to consecutive dehydrogenation on the six-membered rings of cyclohexanones to afford aryl amines. Both experimental and computational studies supported that the α-hydrogen-containing imines derived from primary amines and the enamines from secondary amines follow different reaction pathways. α-Hydrogen-containing imines participate in

reaction via tautomerization to NH-containing enamine tautomers, and the NH-containing enamines kinetically favor both α-radical formation via hydrogen transfer from NH moiety to TEMPO and 1,4-elimination to generate α-amino enones. The enamines from secondary amines, in contrast, prefer β-radical formation via hydrogen abstraction and β-elimination of H/TEMPO to lead to consecutive dehydrogenation owing to relatively low activation barriers. Consequently, due to imine-enamine tautomerization, the α-hydrogen-containing imines are distinctly different in regioselectivity and chemoselectivity from the enamines lacking NH moiety.

These two metal-free reactions both exhibited broad substrate scopes and high functional group tolerance, providing an efficient approach to α-amino-enones, a class of versatile synthetic intermediates for heterocycle syntheses, and offering a valuable complement to the existing metal-catalyzed methods for syntheses of aryl amines. In the α-amino-enone formation reaction, TEMPO has been observed to serve as oxygen transfer reagent. Considering the diverse reactivity of enamine and imine, we anticipate that our findings will arouse chemists' interest in exploration of the reactions involving imine or enamine intermediates based on the difference in reactivity between enamine and tautomerizable imine.

## Methods

**General procedure for α-amino-enone formation reaction**. In a nitrogen-filled glovebox, a 25 mL Schlenk tube equipped with a stir bar was charged with primary amine (0.2 mmol), 2-amino-5-methylbenzoic acid (0.0046 g, 0.03 mmol, 15 mol%), cyclic ketone (0.3 mmol), TEMPO (0.0472 g, 0.3 mmol), molecular sieve (3 Å, 400 mg). The tube was covered with a rubber septum and moved out of the glove box. Then toluene (1.0 mL) was added to the Schlenk tube through the rubber septum using syringes, and then the rubber septum was replaced with a Teflon screwcap under nitrogen flow. The reaction mixture was stirred at 120 °C for 24 h. Upon cooling to room temperature, the mixture was filtered through a pad of silica gel and washed with 10 mL of ethyl acetate. The filtrate was concentrated under reduced pressure and purified by flash chromatography on silica gel to provide the corresponding product.

**General procedure for dehydrogenative aromatization reaction**. In a nitrogen-filled glovebox, a 25 mL Schlenk tube equipped with a stir bar was charged with cyclic ketone (0.2 mmol), secondary amines (0.3 mmol), TE MPO (0.0472 g, 0.3 mmol), molecular sieve (3 Å, 400 mg). The tube was covered with a rubber septum and moved out of the glove box. Then toluene (1.0 mL) was added to the Schlenk tube through the rubber septum using syringes, and then the rubber septum was replaced with a Teflon screwcap under nitrogen flow. The reaction mixture was stirred at 120 °C for 24 h. Upon cooling to room temperature, the mixture was filtered through a pad of silica gel and washed with 10 mL of ethyl acetate. The filtrate was concentrated under reduced pressure and purified by flash chromatography on silica gel to provide the corresponding product.

**Computational details**. All geometry optimizations were carried out with the hybrid density functional theory (DFT) at the level of M06 2X[55] using 6–31 G(d) basis sets[56]. Analytical frequencies were calculated to confirm the correctness of the structure of either a local minimum or a transition state (TS). The solvation effects were considered using the SMD model[57] with toluene as the model solvent. A step size of 0.1 amu$^{1/2}$ bohr was used in the IRC (intrinsic reaction coordinate) procedure[58] to check the connectivity between a transition state and the reactant as well as the product.

The optimized structures were then adopted to calculate the free energies at the level of M062X functional[55] with 6–311 + + G(2d,2p) basis set[59]. The free energy at M062X/6–311 + + G(2d,2p) level in solution phase was calculated according to Eq. 1:[60,61]

$$G_{\text{soln}}^{\text{M062X/6–311++G(2d,2p)}} = E_{\text{gas}}^{\text{M062X/6–311++G(2d,2p)}} + \Delta G_{\text{thermo(soln)}}^{\text{B3LYP/6–31G(d)}} + \Delta G_{\text{solv}} + RT \ln\left(\frac{RT}{P}\right) \quad (1)$$

The first term in the right-hand side is the electronic energy computed at M062X/6-311 + + G(2d,2p) level in gas phase. The second term is the thermal correction to the free energy of the solute in the solution phase at M062X/6-31 G(d) level. The third term is the solvation free energy. The last term denotes the free energy correction from the gas-phase standard state (1 atm) to the solution phase standard state of 1 M. It should be noted that the solvation free energy $\Delta G_{\text{solv}}$ was obtained by using SMD model[57] at the level of B3LYP/6-31 G(d) to make it consistent with the specific methods used in the development of such solvation model[60,61]. All calculations were carried out using the Gaussian 09 program[62].

## Data availability

All the data generated and analysed during this study are included in this article and its Supplementary Information Files, and are also available from the authors on reasonable request. DFT computed Cartesian coordinates of important structures is provided as a separated document in supplementary datasets. Crystallographic data have been deposited at the Cambridge Crystallographic Data Centre (CCDC) as CCDC 1816522 (**3a**), 1816525 (**3ae**), 1816524 (**3ad**) and 1821540 (**7 u**) and can be obtained free of charge from the CCDC via http://www.ccdc.cam.ac.uk/getstructures.

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

## Acknowledgements

This work was supported by the National Key Research and Development Program of China (2017YFA0206801), National Natural Science Foundation of China Grants (21431008, 21332001, u1505242, 21373201 and 21433014), the CAS/SAFEA International Partnership Program for Creative Research Teams, Youth Innovation Promotion Association CAS, the Strategic Priority Research Program of the Chinese Academy of Sciences (XDB20000000 and XDB10040304) and the Key Research Program of the Chinese Academy of Sciences (ZDRW-CN-2016-1).

## Author contributions

X.J. and W.S. conceived the project. X.J. and Y.S. performed major experiments and analysed the data. Z.-N.C. and W.Z. performed computational studies on the mechanism. X.Z. characterised X-ray structures of four compounds. W.S., Y.S., Z.-N.C. and W.Z. prepared the manuscript.

## Additional information

**Competing interests:** The authors declare no competing interests.

