## [Peer Review File · Nature Communications]

Reviewers' comments:

Reviewer #1 (Remarks to the Author):

In the submitted manuscript, Zhuang and Su explore TEMPO reactivity with cyclic ketones and primary or secondary amines. Reaction of a primary amine with cyclohexanone exploits the generation of a tautomerizable imine in situ, resulting in oxygen atom transfer from TEMPO to form α -amino-enones. The reaction of cyclohexanone with a secondary amine affords an enamine intermediate, which subsequently undergoes consecutive dehydrogenation to the aromatized arylamine product. Despite the structural similarities between imines and enamines, the authors propose site-selective H-atom abstraction by TEMPO from these two different species accounts for the different reactivity. The switch in selectivity is intriguing and compelling, and these empirical observations provide the main justification for publication. I find the DFT-based mechanistic analysis to be only marginally compelling, mainly providing specificity for the authors intuitive view of the mechanism (i.e., not really providing a rigorous mechanistic assessment of reaction pathway), and the synthetic impact of this work is marginal (i.e., I don't expect other groups will use this chemistry in synthesis). In the end, I support publication Nature Communications, but not with strong conviction and will defer to the editor to assess the merits in light of other reviews.

Key revisions:

1. The authors are not especially rigorous in defining the reaction stoichiometry. For at least one example from each of the two reaction classes, a rigorous quantification of all reaction (by)products, including the fate of TEMPO species, should be provided. The mechanisms in Figure 2 would seem to implicate a need for more equivalents of TEMPO than are used in the optimized reaction conditions. I may have missed it, but an assessment of the reaction outcome as a function of TEMPO equivalents should be included for each of the two reaction classes..

2. The details provided for DFT calculations are woefully inadequate in the Supporting Information. Whereas experimental details are well documented, a compilation of xyz coordinates for all computed structures should be provided (this content should be standard requirement for all reputable journals). In addition, the authors do little more than provide a computed pathway for their favored mechanism, without any discussion of alternate pathways. This type of presentation does not inspire confidence...
 - a. Computed barriers are not compared to the experimental barriers.
 - b. Free energies for HAT from the N-H of the enamine tautomer are shown in Fig. 2d); how do these compare to the energies for HAT from the C-H of the imine tautomer.

- c. How does the energy and energy barrier of HAT from the beta C-H of the primary amine-derived enamine compare to those of the secondary amine-derived enamine.
- d. TS2A in Fig 2c show a cluster of three water molecules and it show the H-transfer to the oxygen, rather than the nitrogen. None of these features is justified and compared to relevant alternative pathways.

In short, I don't consider the computational presentation suitable for publication. Although much of the requested information may be presented in the SI to avoid distracting in the main article, it needs to be included.

Additional Comments:

1. The authors state 4-methylanthranilic acid is a catalyst for the ketone/amine condensation on line 87, but then state that it impedes the secondary amine condensation on line 137. Further justification should be given for this claim.
2. In line 108, the authors use the successful reactivity with unsubstituted cyclohexanone to conclude that steric hindrance is not a factor in driving the formation of α -amino-enones. However, this statement is not prefaced by any hypothesis or precedent indicating that sterics should play a major role. In this context, 2-substituted cyclohexanones are not demonstrated in any of the synthetic example, nor is there any comment regard the effect of substituents in this position.
3. Specific products discussed in the α -amino-enone scope (starting on line 103) should be clarified with their compound numbers in the appropriate products.
4. There is no optimization table in the text for the dehydrogenative aromatization reaction, and there is no indication in the text that there is an optimization table located in the SI (line 132).
5. Schemes 1 and 2 do not contain any detailed footnote information on how the yields were obtained (GC yields or isolated NMR yields) or indicate different conditions. For example, the reaction to make product 5u in Scheme 2 used 6 eq. of TEMPO, but this is not clarified in the graphic.
6. In line 190, the authors state that HAT by TEMPO "shows some energetically endothermic" when all positions undergoing HAT by TEMPO are endothermic.
7. The authors use a water cluster to assist proton transfer in their DFT calculations, however all reactions are performed in the presence of molecular sieves. This discrepancy should be addressed.

Typos:

1. Line 58: demonstrates should be demonstrate.
2. Lines 59 and 60: taotumerizable should be tautomerizable.

3. The authors say O18-labing instead of labeling. (e.g. line 169)
4. Line 213: "from ambient" is incomplete.
5. There exist other small grammatical errors such as the loss of articles before nouns, incorrect selection of a/an, etc.

Reviewer #2 (Remarks to the Author):

Zhang and Su demonstrated the different chemical selectivity in the reaction of cyclohexones with primary amines and secondary amines in the presence of TEMPO, which provided α -amino-hexenones and anilines product, respectively. The yields for these reactions are generally moderate. Su et al proposed a radical mechanism for these transformation. They further rationalized with a DFT calculation. The authors confirmed that the oxygen atoms in the α -amino-hexenones came from TEMPO after ^{18}O isotope labeled experiment. Over all, I think this manuscript is suitable for publishing on Nature Commun., ONLY IF the following question are addressed clearly and properly.

1. To my understanding, the yields in Scheme 1 are calculated using primary amines as the limiting reagent (e.g. anilines), for example, 3a is 70% yield (30.3 mg out of 0.2 mmol of amine and 0.3 mmol cyclohexanone, page S12). However, the yields in Scheme 2 are based on the cyclohexanones, for example, 5a is 62% yield (40.5 mg out of 0.4 mmol cyclohexanone and 0.6 mmol morpholine, page S32. BTW, there is a typo in this paragraph, it is 0.4 mmol, instead of 0.3 mmol). Could authors rationalize how they take different criteria in one manuscript?

2. Please add the reaction conditions in the Scheme 1 and Scheme 2.

3. In the condition screening section, the authors labeled compound name from 1l, 2l, and 3l at the beginning, this is rare. Please add that molecular sieves does not catalyzed this transformation in the manuscript, as did in the SI (entry 15, page S4).

4. Please add the compounds numbers from line 110 to line 122, e.g., 3b, 3c...

5. They mentioned that molecular sieves promoted the yield, because MS absorbed water that generated from the condensation of ketones and amines. However, in the DFT calculation, authors pointed out that $(\text{H}_2\text{O})_3$ is critical for the hydrogen abstraction. Please rationalize.

6. How about other membered-ring cyclic-ketone substrates for this transformation? Say, the majority of their products are 6-membered cyclic ketones, and one sample for 5-membered ring (3j). How about the 4-membered and 7-membered rings, even larger ones? According to the proposed mechanism, there should be no problems.

7. Please cite references properly. One typical wrong example is ref. 17.

Reviewer #3 (Remarks to the Author):

The authors study in some detail the reactions of two classes of enamines/imines with TEMPO. These are not particularly interesting reactions, and the mechanisms are rather ordinary. The fact that they studied these in great detail on a variety of substrates still does not make them important. Enamines formed from anilines give enamine ketones by oxidation of NH bonds and reaction that looks like alpha oxidation of an imine (but is not). Enamines formed from secondary enamines give allylic oxidation and, eventually, anilines by aromatization. While the authors have done a thorough job of exploring mechanism, including computational studies, this work is most suitable for a good journal like J Organic Chem.

Point-to-point response letter
(Manuscript ID: NCOMMS-18-07096-T, Jie et al.)

Replay to the comments of Reviewer #1

Reviewer #1 Comment 1. *In the submitted manuscript, Zhuang and Su explore TEMPO reactivity with cyclic ketones and primary or secondary amines. Reaction of a primary amine with cyclohexanone exploits the generation of a tautomerizable imine in situ, resulting in oxygen atom transfer from TEMPO to form α -amino-enones. The reaction of cyclohexanone with a secondary amine affords an enamine intermediate, which subsequently undergoes consecutive dehydrogenation to the aromatized arylamine product. Despite the structural similarities between imines and enamines, the authors propose site-selective H-atom abstraction by TEMPO from these two different species accounts for the different reactivity. The switch in selectivity is intriguing and compelling, and these empirical observations provide the main justification for publication.*

Response: We thank the reviewer for his(her) positive comment on our findings.

Reviewer #1 Comment 2. *I find the DFT-based mechanistic analysis to be only marginally compelling, mainly providing specificity for the authors intuitive view of the mechanism (i.e., not really providing a rigorous mechanistic assessment of reaction pathway),*

Response: More comprehensive investigation has been carried out per the reviewer's advice. (1) Kinetics instead of thermodynamics of hydrogen abstraction from all possible sites is now provided to directly evaluate the relative reactivity on different sites. (2) More comprehensive pathways for the elimination of H_A and H_B have been provided. (3) Optimized Cartesian coordinates of important structures in standard xyz file format has been provided. (Please see below for details).

Reviewer #1 Comment 3. *and the synthetic impact of this work is marginal (i.e., I don't expect other groups will use this chemistry in synthesis). In the end, I support publication Nature Communications, but not with strong conviction and will defer to the editor to assess the merits in light of other reviews.*

Response: The reaction of cyclohexanones with and primary amines and TEMPO via

tautomerizable imine intermediates, which is presented herein, offers an efficient approach to α -amino-enones. Due to its dual electronic attitude, α -amino-enone has recently proved to be versatile synthones for facile syntheses of several important classes of heterocycles such as oxazines, azaspirones, quinolinones and quinolones via controlled annulations (*J. Org. Chem.* **2017**, *82*, 7101). α -Amino-enones easily undergo intramolecular oxidative C-C bond formation to construct substituted tetrahydron-1H-carbazol-1-ones (*Tetrahedron*, **2014**, *70*, 2753), a domino reaction with β -bromonitrostyrene to yield pyrroles (*Org. Lett.*, **2010**, *12*, 5281), and intramolecular radical cyclization reaction to form bicycle N-heterocycles (*Tetrahedron*, **1998**, *54*, 13405). Compared with the traditional α -amino-enone syntheses which require a multi-step procedure involving dehydrogenation of cyclohexanone to 2-cyclohexen-1-one, epoxidation of 2-cyclohexen-1-one and subsequent reaction of epoxide with amine, our α -oxygenation reaction of tautomerizable imine represents a straightforward, efficient method for synthesis of α -amino-enone starting from cyclohexanone and primary amine. In view of the versatility of α -amino-enone as a synthon, our direct approach to α -amino-enones will find wide application in organic synthesis.

The other reaction presented by us is the reaction of cyclohexanones with secondary amine and TEMPO to produce aryl amines via consecutive dehydrogenation of enamine intermediate. Aryl amines are ubiquitous building blocks for various organic molecules, electronic materials and pharmaceutical agents. Although aryl amines are generally synthesized using metal-catalyzed cross-coupling of aryl halides with amines, such as Buchwald-Hartwig coupling and Ullmann-type aminations, our method opens up a new avenue to aryl amine synthesis using readily available cyclohexanones as starting materials, and thus provides a concise alternative to the existing methods for syntheses of aryl amines. Importantly, our method would have an impact on retrosynthetic disconnections of the targeted products since starting materials in our method are different from the ones in Buchwald-Hartwig coupling and Ullmann-type aminations.

The broad generality of these two reactions and the importance of their products in organic synthesis should therefore leads to a non-negligible impact on the synthetic chemistry. We have thoroughly discussed the synthetic merits of our two reactions in the revised manuscript. We sincerely hope the reviewer concurs after reading this clarification.

Reviewer #1 Comment 4. *The authors are not especially rigorous in defining the reaction*

stoichiometry. For at least one example from each of the two reaction classes, a rigorous quantification of all reaction (by)products, including the fate of TEMPO species, should be provided. The mechanisms in Figure 2 would seem to implicate a need for more equivalents of TEMPO than are used in the optimized reaction conditions. I may have missed it, but an assessment of the reaction outcome as a function of TEMPO equivalents should be included for each of the two reaction classes.

Response: During screening the reaction parameters for these two reaction classes, we established the reaction stoichiometry including yields of main product and byproducts, and the fate of TEMPO. In response to this suggestion, we have added these data to the results of optimization studies on two model reactions (please see the parts A and B of “Reaction condition Screening “ in Supplementary materials) .

By GC-MS analysis, we observed that TEMPO was converted into the corresponding amine, 2,2,6,6-tetramethylpiperidine (TEMPH), in both classes of reactions. The calculated mechanism for α -amino enone formation reaction (Figure 2d) indicated that generation of one equivalent of α -amino enone consumed two equivalents of TEMPO, in which TEMPO was converted to one equivalent of the corresponding hydroxylamine (TEMPOH) and one equivalent of the corresponding amine (TEMPH). Thus, the TEMPO equivalents implicated in the mechanism appear to be more than 1.5 equivalents of TEMPO used in the optimized conditions. In the calculated mechanism for aryl amine formation (Figure 2c), 4 equivalents of TEMPO were required to abstract H-atom and generate TEMPOH, which appear to be more than 2.5 equivalents used in the optimized conditions. However, in fact, TEMPOH generated from TEMPO is unstable and spontaneously disproportionates to a 2:1 mixture of TEMPO and TEMPH (*Org. Biomol. Chem.* **2003**, *1*, 3232). TEMPO from the disproportionation of TEMPOH would re-participate in reaction, which accounts for the equivalents of TEMPO used in optimized conditions, and the observed conversion of TEMPO to TEMPH in both classes of reactions. To avoid the confusion, we have added the discussion about the re-generation of TEMPO through the disproportionation of TEMPOH in the mechanism part.

We have determined the reaction outcomes with varying TEMPO equivalents, of which results have been added to the revised Supplementary Materials.

Reviewer #1 Comment 5. *Computed barriers are not compared to the experimental barriers.*

Response: We agree that making a direct comparison between the computational and experimental barriers can evaluate the reliability of computed mechanism. From the reaction of TEMPO with cyclohexanone and primary amine, we isolated the α -aminoxylated ketone (please see eq. 3, Scheme 3), which is in agreement with the calculation results that α -site acts as the reactive site for imine molecule B, demonstrating the reliability of computed mechanism.

Reviewer #1 Comment 6. *Free energies for HAT from the N-H of the enamine tautomer are shown in Fig. 2d); how do these compare to the energies for HAT from the C-H of the imine tautomer.*

Response: The Fig. 2a and b have been revised to provide the barriers for hydrogen abstraction with TEMPO from all probable reactive sites, instead of the thermodynamics only. As shown in Fig. 2b, the barriers for hydrogen abstraction from all the C-H sites of the imine tautomer are unfeasible because of the high barriers.

Reviewer #1 Comment 7. *How does the energy and energy barrier of HAT from the beta C-H of the primary amine-derived enamine compare to those of the secondary amine-derived enamine.*

Response: As shown in the revised Fig. 2a and b, hydrogen abstraction from the beta C-H of the primary amine-derived enamine LM1B experiences a barrier of 35.9 kcal/mol, showing a lower activity as compared to hydrogen abstraction from the beta C-H of secondary amine-derived enamine A (33.9 kcal/mol).

Reviewer #1 Comment 8. *TS2A in Fig 2c show a cluster of three water molecules and it show the H-transfer to the oxygen, rather than the nitrogen. None of these features is justified and compared to relevant alternative pathways.*

Response: Actually, both oxygen and nitrogen sites of TEMPO can act as the reactive site for H_B elimination to generate of arylamine species. As shown in Figure S8a (the revised SI), the

pathway for H_B transfer to nitrogen site has been located, showing a slightly lower feasibility as compared to H_B transfer to oxygen site.

Reviewer #1 Comment 9. *In short, I don't consider the computational presentation suitable for publication. Although much of the requested information may be presented in the SI to avoid distracting in the main article, it needs to be included.*

Response: We've largely modified the presentation of the computational part of the manuscript, according to the reviewer's suggestions.

Reviewer #1 Comment 10. *The authors state 4-methylantranilic acid is a catalyst for the ketone/amine condensation on line 87, but then state that it impedes the secondary amine condensation on line 137. Further justification should be given for this claim.*

Response: The 4-methylantranilic acid catalyzed condensation between ketone and primary amine involves the initial condensation of 4-methylantranilic acid with ketone to form an imine intermediate in which ortho carboxyl group interacts with nitrogen atom of imine moiety through intramolecular hydrogen bond to activate C-N double bond towards nucleophilic addition, and the molecule of primary amine is incorporated to this imine intermediate via hydrogen bond between carboxyl group and amino group to form an adduct (*Org. Lett.* 2013, **15**, 1646-1649). Within the newly formed adduct, hydrogen bond enables easy proton-transfer and therefore accelerates addition of primary amine to the C-N double bond of the imine intermediate to produce the targeted imine and release 4-methylantranilic acid. In contrast, due to steric hindrance, relatively bulky secondary amine is difficult to react with the imine intermediate from the condensation of ketone and 4-methylantranilic acid. As a result, 4-methylantranilic acid impedes the condensation of ketone with secondary amine.

We've changed the main text accordingly to include this justification.

Reviewer #1 Comment 11. *In line 108, the authors use the successful reactivity with unsubstituted cyclohexanone to conclude that steric hindrance is not a factor in driving the formation of α -amino-enones. However, this statement is not prefaced by any hypothesis or precedent indicating that sterics should play a major role. In this context, 2-substituted*

cyclohexanones are not demonstrated in any of the synthetic example, nor is there any comment regard the effect of substituents in this position.

Response: In our previous work (*J. Am. Chem. Soc.* **2016**, *138*, 5623.), we established that Cu-catalyzed β -amination of acyclic ketone with a variety of amines via formation of α,β -unsaturated ketone, which are sensitive to steric hindrance. In our model reaction, 4-tert-butyl cyclohexanone was used a substrate to lead to discovery of α -amino-enone formation reaction. To emphasize that the α -amino-enone formation reaction stems from the reactivity of tautomerizable imine rather than steric factor, we gave an example of unsubstituted cyclohexanone in the investigation on substrate scope.

Indeed, 2-substituted cyclohexanones did not participate in this reaction likely due to steric hindrance. In the revised manuscript, we have described this limitation.

Reviewer #1 Comment 12. *Specific products discussed in the α -amino-enone scope (starting on line 103) should be clarified with their compound numbers in the appropriate products.*

Response: We have used compound numbers to describe the products in question for clarity.

Reviewer #1 Comment 13. *There is no optimization table in the text for the dehydrogenative aromatization reaction, and there is no indication in the text that there is an optimization table located in the SI (line 132).*

Response: *Nature Communications* limits the number of Figures and Schemes to no more than six. We therefore put the optimization table for dehydrogenative aromatization in the SI. For clarity, we have indicated this in the main text.

Reviewer #1 Comment 14. Schemes 1 and 2 do not contain any detailed footnote information on how the yields were obtained (GC yields or isolated NMR yields) or indicate different conditions. For example, the reaction to make product 5u in Scheme 2 used 6 eq. of TEMPO, but this is not clarified in the graphic.

Response: We have added the requested detailed footnote information to Schemes 1 and 2, which describes the general conditions for most of products, indicates the procedure to obtain

their yields and highlights the different conditions and procedures for some specific products.

Reviewer #1 Comment 15. *In line 190, the authors state that HAT by TEMPO shows some energetically endothermic when all positions undergoing HAT by TEMPO are endothermic.*

Response: We have changed this statement to “Fig. 2a and 2b demonstrate the free energy changes for abstraction of H-atoms on different positions of imine and enamine by TEMPO.”

Reviewer #1 Comment 16. *The authors use a water cluster to assist proton transfer in their DFT calculations, however all reactions are performed in the presence of molecular sieves. This discrepancy should be addressed.*

Response: We have observed that the selectivity in both α -amino enone formation reaction of imine and dehydrogenative aromatization reaction of enamine **did not** depend on the absence or presence of molecular sieves (please see the product distribution in the parts A and B of “reaction condition screen” in the revised supplementary materials). In both reactions, molecular sieves just improved the yields of products. Since the purpose for the computational studies of the two reaction pathways is to obtain insight into the origin of the switch in selectivity, we did not consider molecular sieves in the computational studies. Considering that water was produced from the condensation of ketones with amines and that trace amount of water from solvents and reagents cannot be precluded, we used water cluster to assist proton transfer in the DFT calculations.

Reviewer #1 Comment 16. Typos: 1. Line 58: demonstrates should be demonstrate. 2. Lines 59 and 60: tautomerizable should be tautomerizable. 3. The authors say O18-labling instead of labeling. (e.g. line 169) 4. Line 213: "from ambient" is incomplete. 5. There exist other small grammatical errors such as the loss of articles before nouns, incorrect selection of a/an, etc.

Response: We've corrected these typos accordingly.

Replay to the comments of Reviewer #2

Reviewer #2 Comment 1. Zhang and Su demonstrated the different chemical selectivity in the reaction of cyclohexones with primary amines and secondary amines in the presence of TEMPO, which provided α -amino-hexenones and anilines product, respectively. The yields for these reactions are generally moderate. Su et al proposed a radical mechanism for these transformation. They further rationalized with a DFT calculation. The authors confirmed that the oxygen atoms in the α -amino-hexenones came from TEMPO after ^{18}O isotope labeled experiment. Over all, I think this manuscript is suitable for publishing on Nature Commun., ONLY IF the following question are addressed clearly and properly.

Response: we appreciate the reviewer's positive comments.

Reviewer #2 Comment 2. To my understanding, the yields in Scheme 1 are calculated using primary amines as the limiting reagent (e.g. anilines), for example, 3a is 70% yield (30.3 mg out of 0.2 mmol of amine and 0.3 mmol cyclohexanone, page S12). However, the yields in Scheme 2 are based on the cyclohexanones, for example, 5a is 62% yield (40.5 mg out of 0.4 mmol cyclohexanone and 0.6 mmol morpholine, page S32. BTW, there is a typo in this paragraph, it is 0.4 mmol, instead of 0.3 mmol). Could authors rationalize how they take different criteria in one manuscript?

Response: We chose the limiting reactant depending on whether the reaction goes to completion. In the reaction of cyclohexanones with primary amines, using primary amines as a limiting reagent allowed the reaction to go to completion while the reaction of limiting cyclohexanones with excessive secondary amines led to complete reaction. Due to the different limiting reagents used in the two reaction classes, we took different criteria for calculation of yields. To avoid this confusion, in the part of description of the second reaction, we explained the reason why we chose different limiting reagents.

Thank the reviewer for his/her pointing out a typo (in page S32), we have corrected it.

Reviewer #2 Comment 3. Please add the reaction conditions in the Scheme 1 and Scheme 2.

Response: We have added the reaction conditions in the detailed footnote information to

Schemes 1 and 2 accordingly.

Reviewer #2 Comment 4. *In the condition screening section, the authors labeled compound name from 1l, 2l, and 3l at the beginning, this is rare. Please add that molecular sieves does not catalyzed this transformation in the manuscript, as did in the SI (entry 15, page S4).*

Response: We have relabeled these compounds with 1a, 2a, and 3a, making numbering the compounds in order. In addition, we have also described that molecular sieves did not catalyze this transformation in the text.

Reviewer #2 Comment 5. *Please add the compounds numbers from line 110 to line 122, e.g., 3b, 3c...*

Response: We have added the compound numbers accordingly to describe the products.

Reviewer #2 Comment 6. *They mentioned that molecular sieves promoted the yield, because MS absorbed water that generated from the condensation of ketones and amines. However, in the DFT calculation, authors pointed out that (H₂O)₃ is critical for the hydrogen abstraction. Please rationalize.*

Response: We have observed that the selectivity in both α -amino enone formation reaction of imine and dehydrogenative aromatization reaction of enamine **did not** depend on the absence or presence of molecular sieves (please see the product distribution in the parts A and B of “reaction condition screen” in the revised supplementary materials). In both reactions, molecular sieves just improved the yields of products. Since the purpose for the computational studies of the two reaction pathways is to obtain insight into the origin of the switch in selectivity, we did not consider molecular sieves in the computational studies. Considering that water was produced from the condensation of ketones with amines and that trace amount of water from solvents and reagents cannot be precluded, we used water cluster to assist proton transfer in the DFT calculations.

Reviewer #2 Comment 7. How about other membered-ring cyclic-ketone substrates for this

transformation? Say, the majority of their products are 6-membered cyclic ketones, and one sample for 5-membered ring (3j). How about the 4-membered and 7-membered rings, even larger ones? According to the proposed mechanism, there should no problems.

Response: For dehydrogenative aromatization of six-membered cyclohexanones with secondary amines, aromatization is the reaction driven force, as a result, it is understandable that this reaction is limited to six-membered ring. When checking the scope for α -amino enone formation reaction. We observed that four-membered or seven-membered cyclic ketone did not work for this reaction. Expanding substrate scope to 4-membered and 7-membered cyclic ketones and identification of the factors to influence the reaction will be the target of our future projects.

Reviewer #2 Comment 7. please cite references properly. One typical wrong example is ref. 17.

Response: Thank the reviewer for pointing out this error. We have corrected it.

Replay to the comments of Reviewer #3

Reviewer #3 comment 1. *The authors study in some detail the reactions of two classes of enamines/imines with TEMPO. These are not particularly interesting reactions, and the mechanisms are rather ordinary. The fact that they studied these in great detail on a variety of substrates still does not make them important. Enamines formed from anilines give enamine ketones by oxidation of NH bonds and reaction that looks like alpha oxidation of an imine (but is not). Enamines formed from secondary enamines give allylic oxidation and, eventually, anilines by aromatization. While the authors have done a thorough job of exploring mechanism, including computational studies, this work is most suitable for a good journal like J Organic Chem.*

Response: We herein, for the first time, discovered that the slight structural difference between the enamine and the enamine tautomer derived from the imine led to the distinctly different chemo- and regioselectivity in their oxidation reactions with TEMPO: the enamines generated from condensation of ketone with secondary amine undergo consecutive dehydrogenation to yield aryl amines while the α -H-containing imines from condensation of

ketones with primary amines undergo α -oxygenation via its enamine tautomer to give α -amino enones. Our discovery changed the conventional concept that the imines containing α -H often exhibit the similar reactivity to enamine due to its rapid tautomerization to enamine tautomer, therefore represents a substantial conceptual advance, which was agreed by the first reviewer (who believes that this mechanism is “intriguing” and “compelling”) and the second reviewer.

In addition, TEMPO, a commonly used reagent, is observed for the first time to serve as an oxygen transfer reagent to the best of our knowledge, and the examples of metal-free dehydrogenative aromatization of enamines are extremely rare. We sincerely hope the reviewer concurs after reading this clarification.

Reviewers' comments:

Reviewer #2 (Remarks to the Author):

Zhang and Su have revised their manuscript per previous reviewer's request. In their new manuscript, they reiterated their observation on different reactivity between the primary and the secondary amines when they react with cyclic ketones in the presence of oxidative radical reagent, TEMPO. The authors further conducted mechanism studies and DFT calculation to rationalize their experimental results: i.e. if these reactions occurred via radical mechanism, the ability to generate more active "NH group-containing" tautomer in the primary amine intermediate (enamine) will mainly provide aniline products; while the secondary amine intermediate (imine), which lack of weak NH bond to generate radical under lower energy barrier, will afford β -amino enone products. However, the authors did not present this conclusion clearly. On the other hand, the yields for these synthetic chemistry are generally moderate. Over all, this manuscript is suitable to publish on Nature Communications.

But, before this manuscript published, authors need do some further modifications.

1. The compound numbers are very confusing. For example, 4-chloro-aniline (1a) in line 86, and 4-tert-butyl-cyclohexanone (2a) in line 87 are not match those in Table 1.
2. The substrate scope section for formation of α -amino enone lines 120 to 140. I am wondering if authors could describe their result in an order of from compound 3a to compound 3ae?
3. In Table 1, should molecular sieves catalyze the reaction? Please provide the control experiments to clarify this question. Let's suppose that the molecule is used solely for absorbing water to promote the reaction and increase yield in these experiments, but authors also claimed that the (H₂O)₃ was critical for the hydrogen abstraction in the TS3B, as shown in Fig. 2d.
4. In lines 120 to 140. Could author please mentioned in the manuscript that 4-membered cyclic ketones is not work for this transformation?

Reviewer #4 (Remarks to the Author):

The C-H bond reactivity of enamine and imine is an important fundamental question for these ubiquitous reactive species. In this work, Su, Zhuang and co-workers discovered that the reaction outcome with TEMPO highly depends on the structure of enamine. With or without the acidic N-H, the degree of oxidation changes significantly. The origins of the switch of selectivity were further elucidated with experimental and computational mechanistic studies. The mechanistic picture is convincing and provides insight for future rational designs of oxidative C-H bond functionalization of enamines. Overall I support its publication in Nature Communications, if my following concerns are resolved.

1. For all the proton shuttle transformations(i.e. TS2A,TS3A,TS3B etc), a bridge of three waters is utilized in the computations. Although I agree that the water bridge is necessary for efficient proton transfer, the number of waters does not have to be three. The authors should provide detailed calculations regarding the number of waters in all proton transfer transition states, in order to present convincing picture of this important elementary step for this work.
2. For the alpha-amino-enone formation reactions with primary amines, most of the transformations were conducted with aromatic amines. However, the aliphatic amine is used in the computations (Figure 2d). The authors should also provide the computational results with a phenyl-substituted imine to address the influence of substitution.
3. I couldn't find any descriptions of the computational methods in the manuscript. This information should be included in the Methods section.
4. Line 230 has a error, aminoxylated ketone (8) should be aminoxylated ketone (11) .

Reviewer #5 (Remarks to the Author):

The manuscript reports a study on the interaction of TEMPO with enamines and imines. This is an interesting topic, which a wide range of applications. The work contains a substantial mechanistic study, both from experimental and computational point of view. I only comment on the computational part, which constitutes my field of expertise.

Calculations, and their interpretation, are below the required standards in 2018 for publication in a journal with an impact factor above 3.0. All the reported free energy profiles in Figure 2 include at least one barrier above 30.0 kcal/mol. This is incompatible with a reaction taking place at room temperature. I admit that there are some trends that agree qualitatively with experiment. But such a poor reproduction of experimental results puts in jeopardy any interpretation from them.

The computational method is substandard, and may be at the origin of the poor agreement with experiment. There is no excuse for carrying out the geometry optimizations in gas phase in this decade. It is also troublesome the absence of dispersion in the geometry optimization, carried out with B3LYP.

Cartesian coordinates of all computed structures should be supplied in the Supporting Information.

Point-by-point responses to comments of reviewers

(Manuscript ID: NCOMMS-18-07096A-Z, by Jie et al.)

1. Responses to the comments of Reviewer #2

Comment 1. Zhang and Su have revised their manuscript per previous reviewer's request. In their new manuscript, they reiterated their observation on different reactivity between the primary and the secondary amines when they react with cyclic ketones in the presence of oxidative radical reagent, TEMPO. The authors further conducted mechanism studies and DFT calculation to rationalize their experimental results: i.e. if these reactions occurred via radical mechanism, the ability to generate more active "NH group-containing" tautomer in the primary amine intermediate (enamine) will mainly provide aniline products; while the secondary amine intermediate (imine), which lack of weak NH bond to generate radical under lower energy barrier, will afford β -amino enone products. However, the authors did not present this conclusion clearly. On the other hand, the yields for these synthetic chemistry are generally moderate. Over all, this manuscript is suitable to publish on Nature Communications.

Response: We thank the reviewer for his/her positive comment and suggestions that are helpful for improvement of our manuscript.

In response to his/her comment on the clear conclusion drawn from the DFT calculation, we have re-organized the paragraph summarizing DFT calculation studies. To this paragraph have been added the sentences "Our calculations demonstrated that the α -hydrogen-containing imines derived from primary amines and the enamines from secondary amines follow different reaction pathways: i) α -hydrogen-containing imines participate in reaction via tautomerization to NH-containing enamine tautomers, and the NH-containing enamines kinetically favor both α -radical formation via hydrogen transfer from NH moiety to TEMPO and 1,4-elimination to preferentially generate α -amino enones; ii) the enamines from secondary amines, in contrast, prefer β -radical formation via hydrogen abstraction and β -elimination of H/TEMPO to lead to consecutive dehydrogenation owing to relatively low activation barriers. Consequently, due to imine-enamine tautomerization, the α -hydrogen-containing imines are distinctly different in regioselectivity and chemoselectivity from the enamines lacking NH moiety." By revision, we expect to present the clear conclusion.

Comment #2. The compound numbers are very confusing. For example, 4-chloro-aniline (1a) in line 86, and 4-tert-butyl-cyclohexanone (2a) in line 87 are not match those in Table 1.

Response: Thank the reviewer for his/her reminding this typo. We have corrected it.

Comment #3. *The substrate scope section for formation of α -amino enone lines 120 to 140. I am wondering if authors could describe their result in an order of from compound 3a to compound 3ae?*

Response: we have agreed with the reviewer and changed the order of describing the results.

Comment #4. *In Table 1, should molecular sieves catalyze the reaction? Please provide the control experiments to clarify this question. Let's suppose that the molecule is used solely for absorbing water to promote the reaction and increase yield in these experiments, but authors also claimed that the $(H_2O)_3$ was critical for the hydrogen abstraction in the TS3B, as shown in Fig. 2d.*

Response: Molecular sieve is not the catalyst for this reaction. The control experiment (Please see entry 6 in Table 1) has been conducted to show that the reaction occurred without molecular sieve. Indeed, molecular sieve enhanced the reaction yields by absorbing water generated in reaction.

In our computational studies of reaction mechanisms, we supposed that water acted as proton transfer assistant to promote the reaction. Actually, we expected to gain the insights into the difference in selectivity between two types of reactions by computational studies. The experiment in entry 6 in Table 1 and the experiment in equation 2 disclosed that the selectivity of these two reactions presented herein did not change in the absence of molecular sieve. Moreover, it is known that the condensation of ketone with amine generates water in the reaction system. Accordingly, the use of water as proton transfer assistant in computational studies of mechanisms would be reasonable.

Comment #5. *In lines 120 to 140. Could author please mentioned in the manuscript that 4-membered cyclic ketones is not work for this transformation?*

Response: we have added in line 141 the sentence "Unfortunately, four-membered cyclic ketone did not work for this reaction."

Responses to the comments of Reviewer #4

Comment 1. *The C-H bond reactivity of enamine and imine is an important fundamental question for these ubiquitous reactive species. In this work, Su, Zhuang and co-workers discovered that the reaction outcome with TEMPO highly depends on the structure of enamine. With or without the acidic N-H, the degree of oxidation changes significantly. The origins of the switch of selectivity were further elucidated with experimental and computational mechanistic studies. The mechanistic picture is*

convincing and provides insight for future rational designs of oxidative C-H bond functionalization of enamines. Overall I support its publication in Nature Communications, if my following concerns are resolved.

Response: We thank the reviewer for his/her positive comment and suggestions that are helpful for improvement of our manuscript.

Comment 2. *For all the proton shuttle transformations (i.e. TS2A, TS3A, TS3B etc), a bridge of three waters is utilized in the computations. Although I agree that the water bridge is necessary for efficient proton transfer, the number of waters does not have to be three. The authors should provide detailed calculations regarding the number of waters in all proton transfer transition states, in order to present convincing picture of this important elementary step for this work.*

Response: According to the calculation results we have just added to the revised supplementary materials, some proton transfer processes feature relatively high activation barriers in the absence of proton-relay chain (see Supplementary Fig. 7), suggesting the necessity of introducing the proper proton-relay chains for rationalization of the reactions. Given that a certain amount of water can be generated from the condensation of ketone with amine and that a trace amount of water may come from reagents, the existence of catalytic amount of water is conceivable. Consequently, water was considered as a possible proton-relay chain here. A cluster made of three water molecules (H₂O)₃ is a general water model in computational investigations (please see references such as *ACS Catal.* **2014**, *4*, 1040; *Inorg. Chem.* **2007**, *46*, 4103; *J. Am. Chem. Soc.* **2007**, *129*, 15503; *J. Comput. Chem.* **2017**, *38*, 438; *Org. Biomol. Chem.* **2013**, *11*, 7923; *Organometallics* **2007**, *26*, 3289). Additionally, (H₂O)₃ cluster was suggested to represent a good compromise between a realistic description and system size for proton exchange processes (please see references: *Catal. Today* **2006**, *115*, 53; *Organometallics* **2006**, *25*, 854). Thus we chose (H₂O)₃ cluster as the water model in our DFT calculations. It is worthy to note that utilization of (H₂O)₃ cluster doesn't mean that the number of water molecules involved in reaction pathways must be three. (H₂O)₃ is just a general water model that describes the proton transfer through proton-relay chain.

In response to this comment, we have discussed the use of (H₂O)₃ cluster as a proton-relay chain in the revised manuscript. To the revised manuscript have been added new references such as *Catal. Today* **115**, 53-60 (2006) and *Organometallics* **25**, 854-861 (2006).

Comment 3. *For the alpha-amino-enone formation reactions with primary amines, most of the transformations were conducted with aromatic amines. However, the aliphatic amine is used in the computations (Figure 2d). The authors should also provide the computational results with a phenyl-substituted imine to address the*

influence of substitution.

Response: We thank the reviewer for this suggestion. Indeed, in the α -amino enone formation reactions, most of examples involved aromatic amines and some examples were the results of aliphatic amines. However, for the consecutive dehydrogenation reactions via enamine, all examples used aliphatic amines as reactants. To disclose the intrinsic difference between enamines and imines, the enamine model molecule **A** and the imine model molecule **B**, which both derive from alkyl amines, were used in our calculation studies. We agree with the reviewer on the influence of substitution. In response to this comment, the reaction of phenyl-substituted imine molecule **C** has been calculated and the results have been provided in revised supplementary materials (see Supplementary Fig. 10). Gratifyingly, the reactions of phenyl- and alkyl-substituted imines follow the similar reaction pathway.

Comment 4. *I couldn't find any descriptions of the computational methods in the manuscript. This information should be included in the Methods section.*

Response: We thank the reviewer for this reminding. In the revised manuscript, the computational methods, which were previously hidden in the supplementary materials, have been moved to the main text.

Comment 5. *Line 230 has a error, aminoxylated ketone (8) should be aminoxylated ketone (11).*

Response: Many thanks for this reminding. We have corrected this error (please see line 272 of the revised manuscript).

Responses to the comments of Reviewer #5

Comment 1. *The manuscript reports a study on the interaction of TEMPO with enamines and imines. This is an interesting topic, which a wide range of applications. The work contains a substantial mechanistic study, both from experimental and computational point of view. I only comment on the computational part, which constitutes my field of expertise.*

Response: We thank the reviewer for his/her positive comment and suggestions that are helpful for improvement of our work.

Comment 2 *Calculations, and their interpretation, are below the required standards in 2018 for publication in a journal with an impact factor above 3.0. All the reported free energy profiles in Figure 2 include at least one barrier above 30.0 kcal/mol. This is incompatible with a reaction taking place at room temperature. I admit that there are some trends that agree qualitatively with experiment. But such a poor*

reproduction of experimental results puts in jeopardy any interpretation from them.

Response: We agree on the reviewer's opinion that the activation barriers above 30.0 kcal/mol are too high for a room temperature reaction. Indeed, during optimization studies of two model reactions, we found that both reactions did not occur at room temperature. However, we thought that this reviewer might misread our reaction conditions probably due to our unclear writing. Our reactions described herein were all conducted at 120 °C.

Comment 3 *The computational method is substandard, and may be at the origin of the poor agreement with experiment. There is no excuse for carrying out the geometry optimizations in gas phase in this decade. It is also troublesome the absence of dispersion in the geometry optimization, carried out with B3LYP.*

Response: Thank the reviewer for these suggestions. We agree with the reviewer's opinion that the solvation effect is important in geometry optimization. In last version of our manuscript, the solvation effect was considered using SMD model in all geometry optimizations. We guessed that our unclear writing in the description of calculation methods caused the reviewer to miss this point.

In response to the reviewer's suggestion that the dispersion interaction needs to be included in geometry optimizations, all geometries have been re-optimized using M062X functional in conjunction with SMD solvation model.

Comment 4 *Cartesian coordinates of all computed structures should be supplied in the Supporting Information.*

Response: Although we provided the optimized Cartesian coordinates in the supplementary materials of last version, we sent these files to the wrong file box in a unsuitable format so that the reviewer had the trouble to find them. We are sorry for that. Now all the optimized Cartesian coordinates have been updated and provided as a separated document in supplementary datasets. These contents have also been specified clearly in the main text.

Reviewer #2 (Remarks to the Author):

Professor Su's feedback is properly addressed my concern. I think this manuscript is now good to accept for publishing on Nat. Commun.

Reviewer #4 (Remarks to the Author):

Publish as is.

The authors have carefully addressed my previous concerns. The additional discussions of the proton-relay mechanism should be helpful to the readers. This work is suitable for publication at current stage.